# Combining Adversarial Guarantees and Stochastic Fast Rates in Online Learning

**Wouter M. Koolen**
Centrum Wiskunde & Informatica
Science Park 123, 1098 XG
Amsterdam, the Netherlands
`wmkoolen@cwi.nl`

**Peter Grünwald**
CWI and Leiden University
`pdg@cwi.nl`

**Tim van Erven**
Leiden University
Niels Bohrweg 1, 2333 CA
Leiden, the Netherlands
`tim@timvanerven.nl`

## Abstract

We consider online learning algorithms that guarantee worst-case regret rates in adversarial environments (so they can be deployed safely and will perform robustly), yet adapt optimally to favorable stochastic environments (so they will perform well in a variety of settings of practical importance). We quantify the friendliness of stochastic environments by means of the well-known Bernstein (a.k.a. generalized Tsybakov margin) condition. For two recent algorithms (Squint for the Hedge setting and MetaGrad for online convex optimization) we show that the particular form of their data-dependent individual-sequence regret guarantees implies that they adapt automatically to the Bernstein parameters of the stochastic environment. We prove that these algorithms attain fast rates in their respective settings both in expectation and with high probability.

## 1 Introduction

We consider online sequential decision problems. We focus on full information settings, encompassing such interaction protocols as online prediction, classification and regression, prediction with expert advice or the Hedge setting, and online convex optimization (see Cesa-Bianchi and Lugosi 2006). The goal of the learner is to choose a sequence of actions with small regret, i.e. such that his cumulative loss is not much larger than the loss of the best fixed action in hindsight. This has to hold even in the worst case, where the environment is controlled by an adversary. After three decades of research there exist many algorithms and analysis techniques for a variety of such settings. For many settings, adversarial regret lower bounds of order $\sqrt{T}$ are known, along with matching individual sequence algorithms [Shalev-Shwartz, 2011].

A more recent line of development is to design adaptive algorithms with regret guarantees that scale with some more refined measure of the complexity of the problem. For the Hedge setting, results of this type have been obtained, amongst others, by Cesa-Bianchi et al. [2007], De Rooij et al. [2014], Gaillard et al. [2014], Sani et al. [2014], Even-Dar et al. [2008], Koolen et al. [2014], Koolen and Van Erven [2015], Luo and Schapire [2015], Wintenberger [2015]. Interestingly, the price for such adaptivity (i.e. the worsening of the worst-case regret bound) is typically extremely small (i.e. a constant factor in the regret bound). For *online convex optimization* (OCO), many different types of adaptivity have been explored, including by Crammer et al. [2009], Duchi et al. [2011], McMahan and Streeter [2010], Hazan and Kale [2010], Chiang et al. [2012], Steinhardt and Liang [2014], Orabona et al. [2015], Van Erven and Koolen [2016].

Here we are interested in the question of whether such adaptive results are strong enough to lead to improved rates in the stochastic case when the data follow a "friendly" distribution. In specific cases it has been shown that fancy guarantees do imply significantly reduced regret. For example Gaillard et al. [2014] present a generic argument showing that a certain kind of second-order regret guarantees

implies constant expected regret (the fastest possible rate) for i.i.d. losses drawn from a distribution with a gap (between expected loss of the best and all other actions). In this paper we significantly extend this result. We show that a variety of individual-sequence second-order regret guarantees imply fast regret rates for distributions under much milder stochastic assumptions. In particular, we will look at the Bernstein condition (see Bartlett and Mendelson 2006), which is the key to fast rates in the batch setting. This condition provides a parametrised interpolation (expressed in terms of the Bernstein exponent $\kappa \in [0, 1]$) between the friendly gap case ($\kappa = 1$) and the stochastic worst case ($\kappa = 0$). We show that appropriate second-order guarantees automatically lead to adaptation to these parameters, for both the Hedge setting and for OCO. In the Hedge setting, we build on the guarantees available for the Squint algorithm [Koolen and Van Erven, 2015] and for OCO we rely on guarantees achieved by MetaGrad [Van Erven and Koolen, 2016]. In both cases we obtain regret rates of order $T^{\frac{1-\kappa}{2-\kappa}}$ (Theorem 2). These rates include the slow worst-case $\sqrt{T}$ regime for $\kappa = 0$ and the fastest (doubly) logarithmic regime for $\kappa = 1$. We show all this, not just in expectation (which is relatively easy), but also with high probability (which is much harder). Our proofs make use of a a convenient novel notation (ESI, for *exponential stochastic inequality*) which allows us to prove such results simultaneously, and which is of independent interest (Definition 5). Our proofs use that, for bounded losses, the Bernstein condition is equivalent to the ESI-Bernstein condition, which we introduce.

The next section introduces the two settings we consider and the individual sequence guarantees we will use in each. It also reviews the stochastic criteria for fast rates and presents our main result. In Section 3 we consider a variety of examples illustrating the breadth of cases that we cover. In Section 4 we introduce ESI and give a high-level overview of our proof.

## 2 Setup and Main Result

### 2.1 Hedge Setting

We start with arguably the simplest setting of online prediction, the Hedge setting popularized by Freund and Schapire [1997]. To be able to illustrate the full reach of our stochastic assumption we will use a minor extension to countably infinitely many actions $k \in \mathbb{N} = \{1, 2, \ldots\}$, customarily called experts. The protocol is as follows. Each round $t$ the learner plays a probability mass function $w_t = (w_t^1, w_t^2, \ldots)$ on experts. Then the environment reveals the losses $\ell_t = (\ell_t^1, \ell_t^2, \ldots)$ of the experts, where each $\ell_t^k \in [0, 1]$. The learner incurs loss $\langle w_t, \ell_t \rangle = \sum_k w_t^k \ell_t^k$. The regret after $T$ rounds compared to expert $k$ is given by

$$R_T^k := \sum_{t=1}^{T} \left( \langle w_t, \ell_t \rangle - \ell_t^k \right).$$

The goal of the learner is to keep the regret small compared to any expert $k$. We will make use of *Squint* by Koolen and Van Erven [2015], a self-tuning algorithm for playing $w_t$. Koolen and Van Erven [2015, Theorem 4] show that Squint with prior probability mass function $\pi = (\pi^1, \pi^2, \ldots)$ guarantees

$$R_T^k \le \sqrt{V_T^k K_T^k} + K_T^k \quad \text{where} \quad K_T^k = O(-\ln \pi^k + \ln \ln T) \qquad \text{for any expert } k. \qquad (1)$$

Here $V_T^k := \sum_{t=1}^{T} \left( \langle w_t, \ell_t \rangle - \ell_t^k \right)^2$ is a second-order term that depends on the algorithm's own predictions $w_t$. It is well-known that with $K$ experts the worst-case lower bound is $\Theta(\sqrt{T \ln K})$ [Cesa-Bianchi and Lugosi, 2006, Theorem 3.7]. Taking a fat-tailed prior $\pi$, for example $\pi^k = \frac{1}{k(k+1)}$, and using $V_T^k \le T$, the above bound implies $R_T^k \le O\left( \sqrt{T(\ln k + \ln \ln T)} \right)$, matching the lower bound in some sense for all $k$ simultaneously.

The question we study in this paper is what becomes of the regret when the sequence of losses $\ell_1, \ell_2, \ldots$ is drawn from some distribution $\mathbb{P}$, not necessarily i.i.d. But before we expand on such stochastic cases, let us first introduce another setting.

### 2.2 Online Convex Optimization (OCO)

We now turn to our second setting called *online convex optimization* [Shalev-Shwartz, 2011]. Here the set of actions is a compact convex set $\mathcal{U} \subseteq \mathbb{R}^d$. Each round $t$ the learner plays a point $w_t \in \mathcal{U}$.

Then the environment reveals a convex loss function $\ell_t : \mathcal{U} \to \mathbb{R}$. The loss of the learner is $\ell_t(w_t)$. The regret after $T$ rounds compared to $u \in \mathcal{U}$ is given by

$$R_T^u := \sum_{t=1}^{T} \left( \ell_t(w_t) - \ell_t(u) \right).$$

The goal is small regret compared to any point $u \in \mathcal{U}$. A common tool in the analysis of algorithms is the linear upper bound on the regret obtained from convexity of $\ell_t$ (at non-differentiable points we may take any sub-gradient)

$$R_T^u \leq \tilde{R}_T^u := \sum_{t=1}^{T} \langle w_t - u, \nabla \ell_t(w_t) \rangle.$$

We will make use of (the full matrix version of) *MetaGrad* by Van Erven and Koolen [2016]. In their Theorem 8, they show that, simultaneously, $\tilde{R}_T^u \leq O\left(DG\sqrt{T}\right)$ and

$$\tilde{R}_T^u \leq \sqrt{V_T^u K_T} + DGK_T \quad \text{where} \quad K_T = O(d \ln T) \qquad \text{for any } u \in \mathcal{U}, \tag{2}$$

where $D$ bounds the two-norm diameter of $\mathcal{U}$, $G$ bounds $\|\nabla \ell_t(w_t)\|_2$ the two-norm of the gradients and $V_T^u := \sum_{t=1}^{T} \langle w_t - u, \nabla \ell_t(w_t) \rangle^2$. The first bound matches the worst-case lower bound. The second bound (2) may be a factor $\sqrt{K_T}$ worse, as $V_T^u \leq G^2 D^2 T$ by Cauchy-Schwarz. Yet in this paper we will show fast rates in certain stochastic settings arising from (2). To simplify notation we will assume from now on that $DG = 1$ (this can always be achieved by scaling the loss).

To talk about stochastic settings we will assume that the sequence $\ell_t$ of loss functions (and hence the gradients $\nabla \ell_t(w_t)$) are drawn from a distribution $\mathbb{P}$, not necessarily i.i.d. This includes the common case of linear regression and classification where $\ell_t(u) = \mathrm{loss}(\langle u, x_t \rangle, y_t)$ with $(x_t, y_t)$ sampled i.i.d. and $\mathrm{loss}$ a fixed one-dimensional convex loss function (e.g. square loss, absolute loss, log loss, hinge loss, ... ).

## 2.3  Parametrised Family of Stochastic Assumptions

We now recall the Bernstein [Bartlett and Mendelson, 2006] stochastic condition. The idea behind this assumption is to control the variance of the excess loss of the actions in the neighborhood of the best action.

We do not require that the losses are i.i.d., nor that the Bayes act is in the model. For the Hedge setting it suffices if there is a fixed expert $k^*$ that is always best, i.e. $\mathbb{E}\left[\ell_t^{k^*} \middle| \mathcal{G}_{t-1}\right] = \inf_k \mathbb{E}\left[\ell_t^k \middle| \mathcal{G}_{t-1}\right]$ almost surely for all $t$. (Here we denote by $\mathcal{G}_{t-1}$ the sigma algebra generated by $\ell_1, \ldots, \ell_{t-1}$, and the *almost surely* quantification refers to the distribution of $\ell_1, \ldots, \ell_{t-1}$.) Similarly, for OCO we assume there is a fixed point $u^* \in \mathcal{U}$ attaining $\min_{u \in \mathcal{U}} \mathbb{E}\left[\ell_t(u) \middle| \mathcal{G}_{t-1}\right]$ at every round $t$. In either case there may be multiple candidate $k^*$ or $u^*$. In the succeeding we assume that one is selected. Note that for i.i.d. losses the existence of a minimiser is not such a strong assumption (if the loss functions $\ell_t$ are continuous, it is even automatic in the OCO case due to compactness of $\mathcal{U}$), while it is very strong beyond i.i.d. Yet it is not impossible (and actually interesting) as we will show by example in Section 3.

Based on the loss minimiser, we define the *excess losses*, a family of random variables indexed by time $t \in \mathbb{N}$ and expert/point $k \in \mathbb{N}/u \in \mathcal{U}$ as follows

$$x_t^k := \ell_t^k - \ell_t^{k^*} \quad \text{(Hedge)} \qquad \text{and} \qquad x_t^u := \langle u - u^*, \nabla \ell_t(u) \rangle \quad \text{(OCO)}. \tag{3}$$

Note that for the Hedge setting we work with the loss directly. For OCO instead we talk about the linear upper bound on the excess loss, for this is the quantity that needs to be controlled to make use of the MetaGrad bound (2). With these variables in place, from this point on the story is the same for Hedge and for OCO. So let us write $\mathcal{F}$ for either the set $\mathbb{N}$ of experts or the set $\mathcal{U}$ of points, and $f^*$ for $k^*$ resp. $u^*$, and let us consider the family $\{x_t^f \mid f \in \mathcal{F}, t \in \mathbb{N}\}$. We call $f \in \mathcal{F}$ *predictors*. With this notation the Bernstein condition is the following.

**Condition 1.** Fix $B \geq 0$ and $\kappa \in [0, 1]$. The family (3) satisfies the $(B, \kappa)$-*Bernstein condition* if

$$\mathbb{E}\left[(x_t^f)^2 \middle| \mathcal{G}_{t-1}\right] \leq B \mathbb{E}\left[x_t^f \middle| \mathcal{G}_{t-1}\right]^\kappa \qquad \text{almost surely for all } f \in \mathcal{F} \text{ and rounds } t \in \mathbb{N}.$$

The point of this stochastic condition is that it implies that the variance in the excess loss gets smaller the closer a predictor gets to the optimum in terms of expected excess loss.

Some authors refer to the $\kappa = 1$ case as the *Massart condition*. Van Erven et al. [2015] have shown that the Bernstein condition is equivalent to the *central condition*, a fast-rate type of condition that has been frequently used (without an explicit name) in density estimation under misspecification. Two more equivalent conditions appear in our proof sketch Section 4. We compare all four formulations in Appendix B.

## 2.4 Main Result

In the stochastic case we evaluate the performance of algorithms by $R_T^{f^*}$, i.e. the regret compared to the predictor $f^*$ with minimal expected loss. The expectation $\mathbb{E}[R_T^{f^*}]$ is sometimes called the *pseudo-regret*. The following result shows that second-order methods automatically adapt to the Bernstein condition. (Proof sketch in Section 4.)

**Theorem 2.** *In any stochastic setting satisfying the $(B, \kappa)$-Bernstein Condition 1, the guarantees* (1) *for Squint and* (2) *for MetaGrad imply fast rates for the respective algorithms both in expectation and with high probability. That is,*

$$\mathbb{E}[R_T^{f^*}] = O\left(K_T^{\frac{1}{2-\kappa}} T^{\frac{1-\kappa}{2-\kappa}}\right),$$

*and for any $\delta > 0$, with probability at least $1 - \delta$,*

$$R_T^{f^*} = O\left((K_T - \ln \delta)^{\frac{1}{2-\kappa}} T^{\frac{1-\kappa}{2-\kappa}}\right),$$

*where for Squint $K_T \coloneqq K_T^{f^*}$ from* (1) *and for MetaGrad $K_T$ is as in* (2).

We see that Squint and MetaGrad adapt automatically to the Bernstein parameters of the distribution, without any tuning. Theorem 2 only uses the form of the second-order bounds and does not depend on the details of the algorithms, so it also applies to any other method with a second-order regret bound. In particular it holds for Adapt-ML-Prod by Gaillard et al. [2014], which guarantees (1) with $K_T = O(\ln|\mathcal{F}| + \ln \ln T)$ for finite sets of experts. Here we focus on Squint as it also applies to infinite sets. Appendix D provides an extension of Theorem 2 that allows using Squint with uncountable $\mathcal{F}$.

Crucially, the bound provided by Theorem 2 is natural, and, in general, the best one can expect. This can be seen from considering the *statistical learning setting*, which is a special case of our setup. Here $(x_t, y_t)$ are i.i.d. $\sim \mathbb{P}$ and $\mathcal{F}$ is a set of functions from $\mathcal{X}$ to a set of predictions $\mathcal{A}$, with $\ell_t^f \coloneqq \ell(y_t, f(x_t))$ for some loss function $\ell : \mathcal{Y} \times \mathcal{A} \to [0, 1]$ such as squared, 0/1, or absolute loss. In this setting one usually considers excess risk, which is the expected loss difference between the learned $\hat{f}$ and the optimal $f^*$. The minimax expected (over training sample $(x^t, y^t)$) risk relative to $f^*$ is of order $T^{-1/2}$ (see e.g. Massart and Nédélec [2006], Audibert [2009]). To get better risk rates, one has to impose further assumptions on $\mathbb{P}$. A standard assumption made in such cases is a Bernstein condition with exponent $\kappa > 0$; see e.g. Koltchinskii [2006], Bartlett and Mendelson [2006], Audibert [2004] or Audibert [2009]; see Van Erven et al. [2015] for how it generalizes the Tsybakov margin and other conditions.

If $\mathcal{F}$ is sufficiently 'simple', e.g. a class with logarithmic entropy numbers (see Appendix D), or, in classification, a VC class, then, if a $\kappa$-Bernstein condition holds, ERM (empirical risk minimization) achieves, in expectation, a better excess risk bound of order $O\left((\log T) \cdot T^{-\frac{1}{2-\kappa}}\right)$. The bound interpolates between $T^{-1/2}$ for $\kappa = 0$ and $T^{-1}$ for $\kappa = 1$ (Massart condition). Results of Tsybakov [2004], Massart and Nédélec [2006], Audibert [2009] suggest that this rate can, in general, not be improved upon, and exactly this rate is achieved by ERM and various other algorithms in various settings by e.g. Tsybakov [2004], Audibert [2004, 2009], Bartlett et al. [2006]. By summing from $t = 1$ to $T$ and using ERM at each $t$ to classify the next data point (so that ERM becomes FTL, follow-the-leader), this suggests that we can achieve a cumulative expected regret $\mathbb{E}[R_T^{f^*}]$ of order $O\left((\log T) \cdot T^{\frac{1-\kappa}{2-\kappa}}\right)$. Theorem 2 shows that this is, indeed, also the rate that Squint attains in such cases if $\mathcal{F}$ is countable and the optimal $f^*$ has positive prior mass $\pi^{f^*} > 0$ (more on this condition below)— we thus see that Squint obtains exactly the rates one would expect from a statistical

learning/classification perspective, and the minimax excess risk results in that setting suggests that these cumulative regret rates cannot be improved in general. It was shown earlier by Audibert [2004] that, when equipped with an oracle to tune the learning rate $\eta$ as a function of $t$, the rates $O\left((\log T) \cdot T^{\frac{1-\kappa}{2-\kappa}}\right)$ can also be achieved by Hedge, but the exact tuning depends on the unknown $\kappa$. Grünwald [2012] provides a means to tune $\eta$ automatically in terms of the data, but his method — like ERM and all algorithms in the references above — may achieve *linear* regret in worst-case settings, whereas Squint keeps the $O(\sqrt{T})$ guarantee for such cases.

Theorem 2 only gives the desired rate for Squint with infinite $\mathcal{F}$ if $\mathcal{F}$ is countable and $\pi^{f^*} > 0$. The combination of these two assumptions is strong or at least unnatural, and OCO cannot be readily used in all such cases either, so in Appendix D we therefore show how to extend Theorem 2 to the case of uncountably infinite $\mathcal{F}$, which can have $\pi^{f^*} = 0$, as long as $\mathcal{F}$ admits sufficiently small entropy numbers. Incidentally, this also allows us to show that Squint achieves regret rate $O\left((\log T) \cdot T^{\frac{1-\kappa}{2-\kappa}}\right)$ when $\mathcal{F} = \bigcup_{i=1,2,\dots} \mathcal{F}_i$ is a countably infinite union of $\mathcal{F}_i$ with appropriate entropy numbers; in such cases there can be, at every sample size, a classifier $\hat{f} \in \mathcal{F}$ with $0$ empirical error, so that ERM/FTL will always over-fit and cannot be used even if the Bernstein condition holds; Squint allows for aggregation of such models. In the remainder of the main text, we concentrate on applications for which Theorem 2 can be used directly, without extensions.

## 3 Examples

We give examples motivating and illustrating the Bernstein condition for the Hedge and OCO settings. Our examples in the Hedge setting will illustrate Bernstein with $\kappa < 1$ and non i.i.d. distributions. Our OCO examples were chosen to be natural and illustrate fast rates without curvature.

### 3.1 Hedge Setting: Gap implies Bernstein with $\kappa = 1$

In the Hedge setting, we say that a distribution $\mathbb{P}$ (not necessarily i.i.d.) of expert losses $\{\ell_t^k \mid t, k \in \mathbb{N}\}$ has *gap* $\alpha > 0$ if there is an expert $k^*$ such that

$$\mathbb{E}\left[\ell_t^{k^*} \big| \mathcal{G}_{t-1}\right] + \alpha \leq \inf_{k \neq k^*} \mathbb{E}\left[\ell_t^k \big| \mathcal{G}_{t-1}\right] \qquad \text{almost surely for each round } t \in \mathbb{N}.$$

It is clear that the condition can only hold for $k^*$ the minimiser of the expected loss.

**Lemma 3.** *A distribution with gap $\alpha$ is $(\frac{1}{\alpha}, 1)$-Bernstein.*

*Proof.* For all $k \neq k^*$ and $t$, we have $\mathbb{E}\left[(x_t^k)^2 \big| \mathcal{G}_{t-1}\right] \leq 1 = \frac{1}{\alpha}\alpha \leq \frac{1}{\alpha} \mathbb{E}\left[x_t^k \big| \mathcal{G}_{t-1}\right].$ □

By Theorem 2 we get the $R_T^{k^*} = O(K_T) = O(\ln \ln T)$ rate. Gaillard et al. [2014] show constant regret for finitely many experts and i.i.d. losses with a gap. Our alternative proof above shows that neither finiteness nor i.i.d. are essential for fast rates in this case.

### 3.2 Hedge Setting: Any $(1, \kappa)$-Bernstein

The next example illustrates that we can sometimes get the fast rates without a gap. And it also shows that we can get any intermediate rate: we construct an example satisfying the Bernstein condition for any $\kappa \in [0, 1]$ of our choosing (such examples occur naturally in classification settings such as those considered in the example in Appendix D).

Fix $\kappa \in [0, 1]$. Each expert $k = 1, 2, \dots$ is parametrised by a real number $\delta_k \in [0, 1/2]$. The only assumption we make is that $\delta_k = 0$ for some $k$, and $\inf_k \{\delta_k \mid \delta_k > 0\} = 0$. For a concrete example let us choose $\delta_1 = 0$ and $\delta_k = 1/k$ for $k = 2, 3, \dots$ Expert $\delta_k$ has loss $1/2 \pm \delta_k$ with probability $\frac{1 \pm \delta_k^{2/\kappa-1}}{2}$ independently between experts and rounds. Expert $\delta_k$ has mean loss $\frac{1}{2} + \delta_k^{2/\kappa}$, and so $\delta_1 = 0$ is best, with loss deterministically equal to $1/2$. The squared excess loss of $\delta_k$ is $\delta_k^2$. So we have the Bernstein condition with exponent $\kappa$ (but no $\kappa' > \kappa$) and constant $1$, and the associated regret rate by Theorem 2.

Note that for $\kappa = 0$ (the hard case) all experts have mean loss equal to $\frac{1}{2}$. So no matter which $k^*$ we designate as the best expert our pseudo-regret $\mathbb{E}[R_T^{k^*}]$ is zero. Yet the experts do not agree, as their losses deviate from $\frac{1}{2}$ independently at random. Hence, by the central limit theorem, with high probability our regret $R_T^{k^*}$ is of order $\sqrt{T}$. On the other side of the spectrum, for $\kappa = 1$ (the best case), we do not find a gap. We still have experts arbitrary close to the best expert in mean, but their expected excess loss squared equals their expected excess loss.

ERM/FTL (and hence all approaches based on it, such as [Bartlett and Mendelson, 2006]) may fail completely on this type of examples. The clearest case is when $\{k \mid \delta_k > \epsilon\}$ is infinite for some $\epsilon > 0$. Then at any $t$ there will be experts that, by chance, incurred their lower loss every round. Picking any of them will result in expected instantaneous regret at least $\epsilon^{2/\kappa}$, leading to linear regret overall.

The requirement $\delta_k = 0$ for some $k$ is essential. If instead $\delta_k > 0$ for all $k$ then there is no best expert in the class. Theorem 19 in Appendix D shows how to deal with this case.

### 3.3 Hedge Setting: Markov Chains

Suppose we model a binary sequence $z_1, z_2, \ldots, z_T$ with $m$-th order Markov chains. As experts we consider all possible functions $f : \{0,1\}^m \to \{0,1\}$ that map a history of length $m$ to a prediction for the next outcome, and the loss of expert $f$ is the 0/1-loss: $\ell_t^f = |f(z_{t-m}, \ldots, z_{t-1}) - z_t|$. (We initialize $z_{1-m} = \ldots = z_0 = 0$.) A uniform prior on this finite set of $2^{2^m}$ experts results in worst-case regret of order $\sqrt{T 2^m}$. Then, if the data are actually generated by an $m$-th order Markov chain with transition probabilities $\mathbb{P}(z_t = 1 \mid (z_{t-m}, \ldots, z_{t-1}) = \boldsymbol{a}) = p_{\boldsymbol{a}}$, we have $f^*(\boldsymbol{a}) = \mathbf{1}\{p_{\boldsymbol{a}} \geq \frac{1}{2}\}$ and

$$\mathbb{E}\left[(x_t^f)^2 \middle| (z_{t-m}, \ldots, z_{t-1}) = \boldsymbol{a}\right] = 1, \qquad \mathbb{E}\left[x_t^f \middle| (z_{t-m}, \ldots, z_{t-1}) = \boldsymbol{a}\right] = 2\left|p_{\boldsymbol{a}} - \frac{1}{2}\right|$$

for any $f$ such that $f(\boldsymbol{a}) \neq f^*(\boldsymbol{a})$. So the Bernstein condition holds with $\kappa = 1$ and $B = \frac{1}{2 \min_{\boldsymbol{a}} |p_{\boldsymbol{a}} - \frac{1}{2}|}$.

### 3.4 OCO: Hinge Loss on the Unit Ball

Let $(x_1, y_1), (x_2, y_2), \ldots$ be classification data, with $y_t \in \{-1, +1\}$ and $x_t \in \mathbb{R}^d$, and consider the *hinge loss* $\ell_t(u) = \max\{0, 1 - y_t\langle x_t, u\rangle\}$. Now suppose, for simplicity, that both $x_t$ and $u$ come from the $d$-dimensional unit Euclidean ball, such that $\langle x_t, u\rangle \in [-1, +1]$ and hence the hinge is never active, i.e. $\ell_t(u) = 1 - y_t\langle x_t, u\rangle$. Then, if the data turn out to be i.i.d. observations from a fixed distribution $\mathbb{P}$, the Bernstein condition holds with $\kappa = 1$ (the proof can be found in Appendix C):

**Lemma 4** (Unregularized Hinge Loss Example). *Consider the hinge loss setting above, where $|\langle x_t, u\rangle| \leq 1$. If the data are i.i.d., then the $(B, \kappa)$-Bernstein condition is satisfied with $\kappa = 1$ and $B = \frac{2\lambda_{max}}{\|\mu\|}$, where $\lambda_{max}$ is the maximum eigenvalue of $\mathbb{E}[xx^\top]$ and $\mu = \mathbb{E}[yx]$, provided that $\|\mu\| > 0$.*

*In particular, if $x_t$ is uniformly distributed on the sphere and $y_t = \mathrm{sign}(\langle \bar{u}, x_t\rangle)$ is the noiseless classification of $x_t$ according to the hyper-plane with normal vector $\bar{u}$, then $B \leq \frac{c}{\sqrt{d}}$ for some absolute constant $c > 0$.*

The excluded case $\|\mu\| = 0$ only happens in the degenerate case that there is nothing to learn, because $\mu = 0$ implies that the expected hinge loss is 1, its maximal value, for all $u$.

### 3.5 OCO: Absolute Loss

Let $\mathcal{U} = [0, 1]$ be the unit interval. Consider the absolute loss $\ell_t(u) = |u - x_t|$ where $x_t \in [0, 1]$ are drawn i.i.d. from $\mathbb{P}$. Let $u^* \in \arg\min_u \mathbb{E}|u - x|$ minimize the expected loss. In this case we may simplify $\langle w - u^*, \nabla \ell(w)\rangle = (w - u^*)\,\mathrm{sign}(w - x)$. To satisfy the Bernstein condition, we therefore want $B$ such that, for all $w \in [0, 1]$,

$$\mathbb{E}\left[\left((w - u^*)\,\mathrm{sign}(w - x)\right)^2\right] \leq B\,\mathbb{E}\left[(w - u^*)\,\mathrm{sign}(w - x)\right]^\kappa.$$

That is,

$$|w - u^*|^{2-\kappa} \leq B 2^\kappa \left|\mathbb{P}(x \leq w) - \frac{1}{2}\right|^\kappa.$$

For instance, if the distribution of $x$ has a strictly positive density $p(x) \geq m > 0$, then $u^*$ is the median and $|\mathbb{P}(x \leq w) - \frac{1}{2}| = |\mathbb{P}(x \leq w) - \mathbb{P}(x \leq u^*)| \geq m|w - u^*|$, so the condition holds with $\kappa = 1$ and $B = \frac{1}{2m}$. Alternatively, for a discrete distribution on two points $a$ and $b$ with probabilities $p$ and $1 - p$, the condition holds with $\kappa = 1$ and $B = \frac{1}{|2p-1|}$, provided that $p \neq \frac{1}{2}$, as can be seen by bounding $|w - u^*| \leq 1$ and $|\mathbb{P}(x \leq w) - \frac{1}{2}| \geq |p - \frac{1}{2}|$.

# 4 Proof Ideas

This section builds up to prove our main result Theorem 2. We first introduce the handy ESI-abbreviation that allows us to reason simultaneously in expectation and with high probability. We then provide two alternative characterizations of the Bernstein condition that are equivalent for bounded losses. Finally, we show how one of these, ESI-Bernstein, combines with individual-sequence second-order regret bounds to give rise to Theorem 2.

## 4.1 Notation: Exponential Stochastic Inequality (ESI, pronounce *easy*)

**Definition 5.** A random variable $X$ is *exponentially stochastically negative*, denoted $X \trianglelefteq 0$, if $\mathbb{E}[e^X] \leq 1$. For any $\eta \geq 0$, we write $X \trianglelefteq_\eta 0$ if $\eta X \trianglelefteq 0$. For any pair of random variables $X$ and $Y$, the *exponential stochastic inequality (ESI)* $X \trianglelefteq_\eta Y$ is defined as expressing $X - Y \trianglelefteq_\eta 0$; $X \trianglelefteq Y$ is defined as $X \trianglelefteq_1 Y$.

**Lemma 6.** *Exponential stochastic negativity/inequality has the following useful properties:*

1. *(Negativity). Let $X \trianglelefteq 0$. As the notation suggests $X$ is negative in expectation and with high probability. That is $\mathbb{E}[X] \leq 0$ and $\mathbb{P}\{X \geq -\ln\delta\} \leq \delta$ for all $\delta > 0$.*

2. *(Convex combination). Let $\{X^f\}_{f \in \mathcal{F}}$ be a family of random variables and let $w$ be a probability distribution on $\mathcal{F}$. If $X^f \trianglelefteq 0$ for all $f$ then $\mathbb{E}_{f \sim w}[X^f] \trianglelefteq 0$.*

3. *(Chain rule). Let $X_1, X_2, \ldots$ be adapted to filtration $\mathcal{G}_1 \subseteq \mathcal{G}_2 \ldots$ (i.e. $X_t$ is $\mathcal{G}_t$-measurable for each $t$). If $X_t | \mathcal{G}_{t-1} \trianglelefteq 0$ almost surely for all $t$, then $\sum_{t=1}^T X_t \trianglelefteq 0$ for all $T \geq 0$.*

*Proof. Negativity*: By Jensen's inequality $\mathbb{E}[X] \leq \ln\mathbb{E}[e^X] \leq 0$, whereas by Markov's inequality $\mathbb{P}\{X \geq -\ln\delta\} = \mathbb{P}\{e^X \geq \frac{1}{\delta}\} \leq \delta\,\mathbb{E}[e^X] \leq \delta$. *Convex combination*: By Jensen's inequality $\mathbb{E}[e^{\mathbb{E}_{f \sim w}[X^f]}] \leq \mathbb{E}_{f \sim w}\mathbb{E}[e^{X^f}] \leq 1$. *Chain rule*: By induction. The base case $T = 0$ holds trivially. For $T > 0$ we have $\mathbb{E}[e^{\sum_{t=1}^T X_t}] = \mathbb{E}[e^{\sum_{t=1}^{T-1} X_t} \mathbb{E}[e^{X_T} | \mathcal{G}_{T-1}]] \leq \mathbb{E}[e^{\sum_{t=1}^{T-1} X_t}] \leq 1$. □

## 4.2 The Bernstein Condition and Second-order Bounds

Our main result Theorem 2, bounds the regret $R_T^{f^*}$ compared to the stochastically optimal predictor $f^*$ when the sequence of losses $\ell_1, \ell_2, \ldots$ comes from a Bernstein distribution $\mathbb{P}$. For simplicity we only consider the OCO setting in this sketch. Full details are in Theorem 11. Our starting point will be the individual-sequence second-order bound (2), which implies $R_T^{f^*} \leq \tilde{R}_T^{f^*} = O(\sqrt{V_T^{f^*} K_T})$. The crucial technical contribution of this paper is to establish that for Bernstein distributions $V_T^{f^*}$ is bounded in terms of $\tilde{R}_T^{f^*}$ with high probability. Combination with the individual-sequence bound then gives that $\tilde{R}_T^{f^*}$ is bounded in terms of a function of itself. And solving the inequality for $\tilde{R}_T^{f^*}$ establishes the fast rates for $R_T^{f^*}$.

To get a first intuition as to why $V_T^{f^*}$ would be bounded in terms of $\tilde{R}_T^{f^*}$, we look at their relation in expectation. Recall that $V_T^{f^*} = \sum_{t=1}^T (x_t^{f_t})^2$ and $\tilde{R}_T^{f^*} = \sum_{t=1}^T x_t^{f_t}$ where $f_t$ is the prediction of the algorithm in round $t$. We will bound $(x_t^{f_t})^2$ in terms of $x_t^{f_t}$ separately for each round $t$. The Bernstein Condition 1 for $\kappa = 1$ directly yields

$$\mathbb{E}[V_T^{f^*}] = \sum_{t=1}^T \mathbb{E}[(x_t^{f_t})^2] \leq B \sum_{t=1}^T \mathbb{E}[x_t^{f_t}] = B\,\mathbb{E}[\tilde{R}_T^{f^*}]. \tag{4}$$

For $\kappa < 1$ the final step of interchanging expectation and sums does not work directly, but we may use $z^\kappa = \kappa^\kappa (1-\kappa)^{1-\kappa} \inf_{\epsilon>0} \{\epsilon^{\kappa-1} z + \epsilon^\kappa\}$ for $z \geq 0$ to rewrite the Bernstein condition as the following set of linear inequalities:

**Condition 7.** The excess loss family (3) satisfies the *linearized $\kappa$-Bernstein condition* if there are constants $c_1, c_2 > 0$ such that we have:

$$c_1 \cdot \epsilon^{1-\kappa} \cdot \mathbb{E}\left[(x_t^f)^2 \big| \mathcal{G}_{t-1}\right] - \mathbb{E}\left[x_t^f \big| \mathcal{G}_{t-1}\right] \;\leq\; c_2 \cdot \epsilon \qquad \text{a.s. for all } \epsilon > 0, f \in \mathcal{F} \text{ and } t \in \mathbb{N}.$$

This gives the following generalization of (4):

$$c_1 \cdot \epsilon^{1-\kappa} \mathbb{E}\left[V_T^{f^*}\right] \;\leq\; \mathbb{E}\left[\tilde{R}_T^{f^*}\right] + c_2 \cdot T \cdot \epsilon. \tag{5}$$

Together with the individual sequence regret bound and optimization of $\epsilon$ this can be used to derive the in-expectation part of Theorem 2.

Getting the in-probability part is more difficult, however, and requires relating $V_T^{f^*}$ and $\tilde{R}_T^{f^*}$ in probability instead of in expectation. Our main technical contribution does exactly this, by showing that the Bernstein condition is in fact equivalent to the following exponential strengthening of Condition 7:

**Condition 8.** The family (3) satisfies the *$\kappa$-ESI-Bernstein condition* if there are $c_1, c_2 > 0$ such that:

$$\left(c_1 \cdot \epsilon^{1-\kappa} \cdot (x_t^f)^2 - x_t^f\right) \mid \mathcal{G}_{t-1} \;\trianglelefteq_{\epsilon^{1-\kappa}}\; c_2 \cdot \epsilon \qquad \text{a.s. for all } \epsilon > 0, f \in \mathcal{F} \text{ and } t \in \mathbb{N}.$$

Condition 8 implies Condition 7 by Jensen's inequality (see Lemma 6 part 1). The surprising converse is proved in Lemma 9 in the appendix. By telescoping over rounds using the chain rule from Lemma 6, we see that ESI-Bernstein implies the following substantial strengthening of (5):

$$c_1 \cdot \epsilon^{1-\kappa} \cdot V_T^{f^*} - \tilde{R}_T^{f^*} \;\trianglelefteq_{\epsilon^{1-\kappa}}\; c_2 \cdot T \cdot \epsilon \qquad \text{a.s. for all } \epsilon > 0, T \in \mathbb{N}. \tag{6}$$

Now the second-order regret bound (2) can be rewritten, using $2\sqrt{ab} = \inf_\gamma \gamma a + b/\gamma$, as:

$$\text{for every } \gamma > 0: \; 2\tilde{R}_T^{f^*} \;\leq\; 2\sqrt{V_T^{f^*} \cdot K_T} + 2K_T \;\leq\; \gamma \cdot V_T^{f^*} + \frac{K_T}{\gamma} + 2K_T.$$

Plugging in $\gamma = c_1 \epsilon^{1-\kappa}$ we can chain this inequality with (6) to give, for all $\epsilon > 0$,

$$2\tilde{R}_T^{f^*} \;\trianglelefteq_{\epsilon^{1-\kappa}}\; \tilde{R}_T^{f^*} + c_2 \cdot T \cdot \epsilon + \frac{K_T}{c_1 \cdot \epsilon^{1-\kappa}} + 2K_T, \tag{7}$$

and both parts of Theorem 2 now follow by rearranging, plugging in the minimiser $\epsilon \asymp K_T^{\frac{1}{2-\kappa}} T^{\frac{1-\kappa}{2-\kappa}}$, and using Lemma 6 part 1.

### Acknowledgments

Koolen acknowledges support by the Netherlands Organization for Scientific Research (NWO, Veni grant 639.021.439).

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
