[Supplementary Material]

# A  Proof of Main Result

In this section we prove our main result, Theorem 2. As stated in Section 4, the technically challenging part is to show that the Bernstein condition implies the ESI-Bernstein condition. This is done implicitly in Section A.1, Lemma 9, which in fact proves a more general statement about the *normalized cumulant generating function*. Lemma 9 is then used in the proof of Theorem 11 to give a regret bound that holds in the ESI-sense and that, unlike the original Squint and MetaGrad bounds, does not refer to the quantity $V_T^{f^*}$ which depends on the predictions of the algorithms themselves. This is a generalization of the bound (7) in Section 4.

We then use Theorem 11 to prove the main theorem, Theorem 2.

## A.1  Normalized Cumulant Generating Function

Consider the family (3) of excess loss variables $x_t^f$. We assume that $x_t^f \in [-1, 1]$ is bounded in a range of width 2 and has positive mean $\mathbb{E}[x_t^f | \mathcal{G}_{t-1}] \geq 0$ by definition of $f^*$. Consider the *normalized cumulant generating function* for $\eta \geq 0$:

$$\epsilon_t^f(\eta) \coloneqq \frac{1}{\eta} \ln \mathbb{E}\left[e^{-\eta x_t^f} \big| \mathcal{G}_{t-1}\right]$$

By construction

$$-x_t^f \trianglelefteq_\eta \epsilon_t^f(\eta). \tag{8}$$

Boundedness of $x_t^f \in [-1, 1]$ immediately results in $\epsilon_t^f(\eta) \in [-1, 1]$. Moreover, Hoeffding's inequality (see e.g. Cesa-Bianchi and Lugosi [2006, Lemma 2.2]) tells us that $\epsilon_t^f(\eta) \leq \eta/2$ while Jensen's inequality gives $\epsilon_t^f(\eta) \geq -\mathbb{E}\left[x_t^f \big| \mathcal{G}_{t-1}\right]$. The dual representation $\epsilon_t^f(\eta) = \sup_Q -\mathbb{E}_Q[x] - \frac{1}{\eta} \mathrm{KL}\big(Q(x) \big\| \mathbb{P}(x_t^f | \mathcal{G}_{t-1})\big)$ reveals that $\epsilon_t^f(\eta)$ is increasing in $\eta$. The value at $\eta = 0$ is obtained by continuity from $\epsilon_t^f(0) \coloneqq \lim_{\eta \to 0} \epsilon_t^f(\eta) = -\mathbb{E}[x_t^f | \mathcal{G}_{t-1}] \leq 0$.

To get a uniform control over the class $\mathcal{F}$, we will make use of the maximum

$$\epsilon_t(\eta) \coloneqq \sup_{f \in \mathcal{F}} \epsilon_t^f(\eta) \qquad \text{and} \qquad \epsilon(\eta) \coloneqq \sup_t \epsilon_t(\eta). \tag{9}$$

The functions $\epsilon_t$ and $\epsilon$ inherit most properties of each $\epsilon_t^f$, but in addition since $f^* \in \mathcal{F}$ and $\epsilon_t^{f^*}(\eta) = 0$, we see that $\epsilon_t(\eta) \geq 0$ and also that $\epsilon_t(0) = 0$. Moreover, since $\epsilon_t(\eta) \leq \eta/2$ we have $\lim_{\eta \to 0} \epsilon_t(\eta) = 0$.

**Lemma 9.  [Main Lemma]** *For any random variables $x_1^f, x_2^f, \ldots$ as above, we have:*

$$\left(c\eta \cdot (x_t^f)^2 - x_t^f\right) | \mathcal{G}_{t-1} \trianglelefteq_\eta \epsilon(2\eta) + c\eta\epsilon(2\eta)^2 \qquad \text{a.s. for all } \eta \geq 0, \ f \in \mathcal{F} \text{ and rounds } t \in \mathbb{N}, \tag{10}$$

*where $c = \frac{1}{1+\sqrt{1+4\eta^2}}$.*

Section A.4 is dedicated to the proof. Note that, for small $\eta$, the right-hand side is $O(\epsilon(2\eta))$. Comparing to (8), we see that the lemma then implies the perhaps surprising fact that we can add a quadratic to the left-hand side of (8) and yet the same inequality holds up to constant factors.

To give explicit meaning to Lemma 9 we need to get good bounds on $\epsilon(2\eta)$. The following proposition, a special case of [Van Erven et al., 2015, Theorem 5.4, Part 1], implies such bounds under a Bernstein condition.

**Proposition 10.** *Suppose that for all $f$, $t$, $x_t^f \in [-1, 1]$ a.s., and that a $(B, \kappa)$-Bernstein condition holds for some $\kappa \in [0, 1]$. For $0 \leq \eta \leq 1.79328$, we have:*

$$\epsilon(\eta) \leq \frac{1-\kappa}{\kappa}(B\eta\kappa)^{\frac{1}{1-\kappa}} \leq (B\eta)^{\frac{1}{1-\kappa}}. \tag{11}$$

We note that the bound resulting from combining Proposition 10 with Lemma 9, for the special case $\kappa = 1$, would also follow (up to constant factors) from [Gaillard et al., 2014, Theorem 12]. Yet, their result cannot be straightforwardly adapted to the case $\kappa < 1$.

*Proof.* It is sufficient to show that any random variable $x$ on $[-1, 1]$ with $\mathbb{E}[x^2] \le B(\mathbb{E}[x])^\kappa$ for some $0 < \kappa \le 1$ must satisfy

$$\frac{1}{\eta} \ln \mathbb{E}\left[e^{-\eta x}\right] \le \frac{1-\kappa}{\kappa}(B\eta\kappa)^{\frac{1}{1-\kappa}}.$$

By the Bernstein Sandwich ([Koolen et al., 2014, Lemma C.2], or, for a perhaps easier proof, [Van Erven et al., 2015, Lemma 5.6]), simplifying $e^\eta - \eta - 1 \le \eta^2$, which holds for small enough $\eta \le 1.79328$, we have:

$$\frac{1}{\eta}\ln\mathbb{E}\left[e^{-\eta x}\right] \le \eta\mathbb{E}[(x)^2] - \mathbb{E}[x] \le \eta B\mathbb{E}[x]^\kappa - \mathbb{E}[x] \le \sup_{x \ge 0}\left\{\eta Bx^\kappa - x\right\} = \frac{1-\kappa}{\kappa}(B\eta\kappa)^{\frac{1}{1-\kappa}},$$

where the maximizer is found at $x = (B\eta\kappa)^{\frac{1}{1-\kappa}}$ (which is $\le 1$ when $B\kappa\eta \le 1$, so for small enough $\eta$ this is a reasonable point). The last inequality of the claim follows from the fact that $\kappa \mapsto \frac{1-\kappa}{\kappa}\kappa^{\frac{1}{1-\kappa}}$ is decreasing with value 1 at $\kappa = 0$. $\qquad\square$

In the next subsection, in Theorem 11, we directly use Lemma 9 to prove regret bounds for Squint and MetaGrad in terms of the underlying normalized cumulant generating function $\epsilon(\eta)$. In Section A.3 we then prove our main result, Theorem 2, by plugging in the bound on $\epsilon(\eta)$ provided by Proposition 10. This is a strictly more general approach than suggested in the proof outline of Section 4. There, we (a) claimed that the Bernstein condition implies an ESI-Bernstein condition, and then, (b), in Eq. (7), we presented an analogue of Theorem 11 below that only works under this ESI-Bernstein condition. To see how claim (a) also follows from Lemma 9, plug in $\eta \asymp \epsilon^{1-\kappa}$ and apply Proposition 10.

## A.2 Regret Bound in terms of Normalized Cumulant

We now state and prove Theorem 11, which transforms the Squint and MetaGrad bounds — that involve the algorithm itself on the right-hand side — into stochastic bounds that depend instead on the normalized cumulant generating function; it generalizes Eq. (7) in Section 4. The proof relies on Lemma 9.

**Theorem 11.** *Consider either Squint in the Hedge setting or MetaGrad for OCO. Let $\left\{x_t^f \middle| f \in \mathcal{F}\right\}$ be the associated excess loss family from (3), and let $\epsilon(\eta)$ be, as in (9), the corresponding maximal normalized cumulant generating function. For the Hedge setting let $K_T := K_T^{f^*}$ as in (1), for OCO let $K_T$ be as in (2). Then for each $\gamma \ge 0$ with $c = \frac{1}{1+\sqrt{1+4\gamma^2}}$,*

$$R_T^{f^*} \trianglelefteq_\gamma \frac{K_T}{c\gamma} + T\epsilon(2\gamma)(1 + c\gamma^2) + 2K_T.$$

*Proof.* For the Hedge setting, let us write $x_t := \mathbb{E}_{k \sim w_t}\left[x_t^k\right]$ for the excess loss of the learner in round $t$. Then $x_t \in [-1, 1]$ and $-x_t \mid \mathcal{G}_{t-1} \trianglelefteq_\eta \epsilon(\eta)$ a.s. by Lemma 6 part 2. Now by definition of $x_t^k$ in (3) and $R_T^k$ and $V_T^k$ (see Section 2.1)

$$\sum_{t=1}^T x_t = \sum_{t=1}^T\left(\langle w_t, \ell_t\rangle - \ell_t^{k^*}\right) = R_T^{k^*} \quad\text{and}\quad \sum_{t=1}^T(x_t)^2 = \sum_{t=1}^T\left(\langle w_t, \ell_t\rangle - \ell_t^{k^*}\right)^2 = V_T^{k^*}.$$

For OCO, let us write $x_t := x_t^{w_t}$ for the excess loss of the learner in round $t$. Again $x_t \in [-1, 1]$ and we have $(-x_t \mid \mathcal{G}_{t-1}) \trianglelefteq_\eta \epsilon(\eta)$ by construction of $\epsilon(\eta)$. Moreover, from the definition of $\tilde{R}_T^u$ and $V_T^u$ from Section 2.2,

$$\sum_{t=1}^T x_t = \sum_{t=1}^T\langle w_t - u^*, \nabla\ell_t(w_t)\rangle = \tilde{R}_T^{u^*} \quad\text{and}\quad \sum_{t=1}^T(x_t)^2 = \sum_{t=1}^T\langle w_t - u^*, \nabla\ell_t(w_t)\rangle^2 = V_T^{u^*}.$$

With this notation the second-order regret bounds (1) and (2) both state

$$\sum_{t=1}^T x_t \le \sqrt{\left(\sum_{t=1}^T x_t^2\right)K_T} + K_T. \tag{12}$$

Now fix $\gamma \geq 0$. For any $t$, as $-x_t \trianglelefteq_\eta \epsilon(\eta)$, Lemma 9 and Hoeffding's Inequality $\epsilon(2\gamma) \leq \gamma$ give

$$c\gamma x_t^2 - x_t \trianglelefteq_\gamma \epsilon(2\gamma)(1 + c\gamma^2).$$

By telescoping over rounds (using the chain rule Lemma 6 part 3), we obtain

$$c\gamma \sum_{t=1}^T x_t^2 - \sum_{t=1}^T x_t \trianglelefteq_\gamma T\epsilon(2\gamma)(1 + c\gamma^2). \tag{13}$$

The individual sequence regret bound (12) gives us (since $2\sqrt{ab} = \inf_\eta \eta a + b/\eta$) for every $\eta \geq 0$

$$\sum_{t=1}^T x_t \leq \frac{\eta}{2} \sum_{t=1}^T x_t^2 + \frac{K_T}{2\eta} + K_T.$$

Plugging in $\eta = c\gamma$ (this implies $\eta \in [0, 1/2]$ and $\gamma = \frac{2\eta}{1-4\eta^2}$) and combination with (13) results in

$$\sum_{t=1}^T x_t \trianglelefteq_\gamma \frac{K_T}{c\gamma} + T\epsilon(2\gamma)(1 + c\gamma^2) + 2K_T.$$

For the Hedge setting this proves the theorem. For OCO we finish with $R_T^{u^*} \leq \tilde{R}_T^{u^*}$. □

## A.3 Proof of Theorem 2

We are now ready to prove Theorem 2 based on Theorem 11 and Proposition 10.

*Proof.* By Proposition 10, the $(B, \kappa)$-Bernstein condition implies that the normalized cumulant generating function $\epsilon(\eta)$ satisfies $\epsilon(\eta) \leq (\eta B)^{\frac{1}{1-\kappa}}$. By Theorem 11, using $1/c \leq 2(1 + \gamma^2)$ and $c \leq \frac{1}{2}$, we find that for all $\gamma \geq 0$

$$R_T^{f^*} \trianglelefteq_\gamma (1 + \gamma^2)\frac{2K_T}{\gamma} + (1 + \frac{1}{2}\gamma^2)\epsilon(2\gamma)T + 2K_T. \tag{14}$$

By Lemma 6 part 1 this implies for all $\gamma \geq 0$

$$\mathbb{E}[R_T^{f^*}] \leq (1 + \gamma^2)\frac{2K_T}{\gamma} + (1 + \frac{1}{2}\gamma^2)\epsilon(2\gamma)T + 2K_T.$$

It remains to tune $\gamma$ to exploit the stochastic condition expressed by $\epsilon$. Reducing the above right-hand-side expression to its main terms and setting the derivative to zero suggests picking

$$\hat{\gamma} = \left(\frac{2K_T(1-\kappa)(2B)^{-\frac{1}{1-\kappa}}}{T}\right)^{\frac{1-\kappa}{2-\kappa}} = O\left((K_T/T)^{\frac{1-\kappa}{2-\kappa}}\right).$$

For $T$ large enough such that $2\hat{\gamma} \leq 1.79$, we find

$$\mathbb{E}[R_T^{f^*}] \leq (2 - \kappa)(4K_T B)^{\frac{1}{2-\kappa}}(T/(1-\kappa))^{\frac{1-\kappa}{2-\kappa}} + (5 - \kappa)K_T$$

Finally, using that $(2 - \kappa)(4B)^{\frac{1}{2-\kappa}}(1/(1-\kappa))^{\frac{1-\kappa}{2-\kappa}}$ is maximized in $\kappa$ at $\kappa = 1 - \frac{1}{4B}$ where it takes value $1 + 4B$, we may simplify this to

$$\mathbb{E}[R_T^{f^*}] \leq (1 + 4B)(K_T/4)^{\frac{1}{2-\kappa}}T^{\frac{1-\kappa}{2-\kappa}} + (5 - \kappa)K_T = O\left(K_T^{\frac{1}{2-\kappa}}T^{\frac{1-\kappa}{2-\kappa}}\right),$$

which gives the first claim of the Theorem. Lemma 6 applied to (14) also implies that for all $\delta \geq 0$ with probability at least $1 - \delta$

$$R_T^{f^*} \leq (1 + \gamma^2)\frac{2K_T}{\gamma} + (1 + \frac{1}{2}\gamma^2)\epsilon(2\gamma)T + 2K_T + \frac{-\ln\delta}{\gamma}.$$

Tuning $\gamma$ as before with $K_T$ replaced by $K_T + \frac{-\ln\delta}{2}$ yields the second claim. □

## A.4 Proof of Lemma 9

In this proof we consider any random variable $x \in [-1, 1]$ and denote its normalized cumulant generating function by $\epsilon(\eta) = \frac{1}{\eta} \ln \mathbb{E}\left[e^{-\eta x}\right]$. (In particular, see Definition 5, $-x \trianglelefteq_\eta \epsilon(\eta)$ for all $\eta \geq 0$.) Thus, there is neither a time $t$ nor a function $f$ in the definition of $x$ or $\epsilon$.

We will show that for any $\eta \geq 0$

$$\frac{1}{\eta} \ln \mathbb{E}\left[e^{c\eta^2 x^2 - \eta x}\right] \;\leq\; \epsilon(2\eta) + c\eta\epsilon(2\eta)^2 \qquad \text{where} \qquad c = \frac{1}{1 + \sqrt{1 + 4\eta^2}}. \tag{15}$$

Noting that the right-hand side is an increasing function in $\epsilon(2\eta)$ (the quadratic in $\epsilon(2\eta)$ has positive derivative for all $\epsilon(2\eta) \geq -\frac{1+\sqrt{4\eta^2+1}}{2\eta} < -1$) we can apply this equation sequentially to the random variables $x_t^f \mid \mathcal{G}_{t-1}$ giving (with $\epsilon(\eta)$ now defined as $\sup_t \epsilon_t(\eta)$ as in (9), the desired result (10).

Below we will prove Theorem 14, which is actually stronger than (15). Applying Theorem 14 with $\gamma = 2\eta$ and the largest admissible $c$ for (16) gives us (15).

We would like to remark that our solution to this problem was inspired by the general moments problem studied by Mehta and Williamson [2014, Section 3], especially because this connection became invisible during the simplification of our proofs.

We will be thinking about two learning rates, $0 \leq \eta \leq \gamma$. The larger one, $\gamma$, will be where we evaluate $\epsilon(\gamma)$. So $\gamma$ controls the strength of the assumption. The smaller one, $\eta$, will be the learning rate at which we obtain the conclusion. The point is to get a large amount of quadratic $x^2$ in the conclusion, as governed by the constant $c$. Obviously, the more greedy we are in $\eta$ and $\gamma$, the smaller the $c$ for which we can get any traction. This trade-off is captured by the following relationship between $\gamma$, $\eta$ and $c$ that we will make use of throughout this section.

$$0 \;\leq\; c \;\leq\; \frac{\sqrt{2|2\eta - \gamma| + \gamma^2 + 1} - |2\eta - \gamma| - 1}{4\eta^2} \tag{16}$$

Positivity of $c$ is not that important, as the desired inequality is trivial for $c \leq 0$. The following inequality is useful later.

**Lemma 12.** *Let $0 \leq \eta \leq \gamma$ and $c$ satisfy* (16)*. Then*

$$1 \;\geq\; 2c\eta$$

*Proof.* We need to show

$$2\eta \;\geq\; \sqrt{2|2\eta - \gamma| + \gamma^2 + 1} - |2\eta - \gamma| - 1$$

that is

$$\left(2\eta + |2\eta - \gamma| + 1\right)^2 \;\geq\; 2|2\eta - \gamma| + \gamma^2 + 1$$

Expanding the left-hand side square results in

$$4\eta^2 + 4\eta|2\eta - \gamma| + 4\eta + |2\eta - \gamma|^2 + 2|2\eta - \gamma| + 1 \;=\; 4\eta(2\eta - \gamma) + 4\eta|2\eta - \gamma| + 4\eta + \gamma^2 + 2|2\eta - \gamma| + 1$$

which definitely exceeds the right-hand side above. $\qquad\square$

We now put our assumption to use. In the following Lemma we show that it implies a not-in-expectation-but-with-a-correction-term version of the result we are after.

**Lemma 13.** *Fix $0 \leq \eta \leq \gamma$ and let $c$ satisfy* (16)*. Then for each $x \in [-1, 1]$ and $\epsilon \in [-1, 1]$ we have*

$$e^{c\eta^2 x^2 - \eta x} - \frac{e^{-\gamma(x+\epsilon)} - 1}{\gamma}\eta(1 + 2c\eta\epsilon)e^{c\eta^2 \epsilon^2 + \eta\epsilon} \;\leq\; e^{c\eta^2 \epsilon^2 + \eta\epsilon}.$$

*Proof.* We will show that the left-hand side is maximized over $x \in [-1, 1]$ at $x = -\epsilon$. First, its derivative equals

$$e^{-\gamma x}\eta\big(h(-\epsilon) - h(x)\big) \qquad \text{where} \qquad h(x) = (1 - 2c\eta x)e^{c\eta^2 x^2 + (\gamma - \eta)x}.$$

This indeed equals zero at $x = -\epsilon$. To show that $x = -\epsilon$ is indeed a maximum and that there are no other maxima it suffices to show that $h(x)$ is increasing on $x \in [-1, 1]$. We have

$$h'(x) = \left(-4c^2\eta^3 x^2 + 2c\eta x(2\eta - \gamma) - 2c\eta + \gamma - \eta\right)e^{c\eta^2 x^2 + (\gamma - \eta)x}$$

As the term in parentheses is concave in $x$, it suffices to show that $h'(x) \geq 0$ for $x \in \{-1, 1\}$, i.e.

$$-4c^2\eta^3 - 2c\eta|2\eta - \gamma| - 2c\eta + \gamma - \eta \geq 0$$

Solving the quadratic in $c$, we see that this holds if (16), as required. $\qquad\square$

Finally, we are ready for the general version of the claim.

**Theorem 14.** *Pick $0 \leq \eta \leq \gamma$ and $c$ satisfying* (16). *Let $\epsilon \in [-1, 1]$. Then for any $x \in [-1, 1]$ with $\mathbb{E}\,e^{-\gamma x} \leq e^{\gamma\epsilon}$ we have*

$$\mathbb{E}\,e^{c\eta^2 x^2 - \eta x} \leq e^{c\eta^2 \epsilon^2 + \eta\epsilon}.$$

*Proof.* Taking expectation over Lemma 13, we find

$$\mathbb{E}\,e^{c\eta^2 x^2 - \eta x} \leq e^{c\eta^2 \epsilon^2 + \eta\epsilon} + \frac{\mathbb{E}\,e^{-\gamma(x+\epsilon)} - 1}{\gamma}\eta(1 + 2c\eta\epsilon)e^{c\eta^2\epsilon^2 + \eta\epsilon},$$

and the claim follows by bounding the right-most term by 0. (Note that the factor $1 + 2c\eta\epsilon$ is positive by Lemma 12.) $\qquad\square$

# B  Four Equivalent Versions of the Bernstein Condition

In earlier work [Van Erven et al., 2015], we provided the *central condition*, which, for bounded loss functions, is essentially equivalent to the Bernstein condition. The central condition provides the link between Bernstein and several other fast-rate conditions in the literature such as the condition needed for fast rates in density estimation under misspecification; the Juditsky-Rigollet-Tsybakov condition, and several others — see [Van Erven et al., 2015] for details. In Section 4 we provided two other equivalent reformulations of the Bernstein condition. In this appendix we contrast these four conditions and explain how the central condition implicitly does play a role in our results.

It is instructive survey the conditions starting from a slight generalization of the Bernstein condition as in [Van Erven et al., 2015], in which the relation between second and first moment of $x_t^t$ can be any nondecreasing function $\nu : \mathbb{R}_0^+ \to \mathbb{R}_0^+$ which satisfies that $\nu(y)/y$ is non-increasing in $y$. Although the special case with $\nu(y) = y^\kappa$ for $\kappa \in [0, 1]$ gives the original Bernstein condition, and remains the most important, all the results in this paper readily generalize to the generalized form. For simplicity, in the definitions below we allow the constant factors (that do not affect the rates) to be arbitrary. As can be seen from Condition 1, the generalized Bernstein condition then reads as:

**Condition 15.** Fix a function $\nu$ as above. The family (3) satisfies the *$\nu$-Bernstein condition* if there is $c > 0$ such that

$$c \cdot \mathbb{E}\left[(x_t^f)^2\big|\mathcal{G}_{t-1}\right] - \nu\left(\mathbb{E}\left[x_t^f\big|\mathcal{G}_{t-1}\right]\right) \leq 0 \qquad \text{a.s. for all } f \in \mathcal{F} \text{ and rounds } t \in \mathbb{N}.$$

The second condition we encountered was the *linearized* Bernstein condition, which in our generalized form becomes:

**Condition 16.** The family (3) satisfies the *linearized $\nu$-Bernstein condition* if there are constants $c_1, c_2 > 0$ such that, for the (non-decreasing) function $\eta_\epsilon := \frac{\epsilon}{\nu(\epsilon)}$, we have:

$$c_1 \cdot \eta_\epsilon \cdot \mathbb{E}\left[(x_t^f)^2\big|\mathcal{G}_{t-1}\right] - \mathbb{E}\left[x_t^f\big|\mathcal{G}_{t-1}\right] \leq c_2 \cdot \epsilon \qquad \text{a.s. for all } \epsilon \geq 0, f \in \mathcal{F} \text{ and } t \in \mathbb{N},$$

where we employ the convention $\eta_0 = \inf_{\epsilon > 0} \epsilon/\nu(\epsilon)$.

Then we encountered the *$\nu$-ESI-Bernstein condition*:

**Condition 17.** Let $\nu$ be a function as above and $\eta_\epsilon$ be defined as above. The family (3) satisfies the *$\nu$-ESI-Bernstein condition* if there are $c_1, c_2 > 0$ such that:

$$\left(c_1 \cdot \eta_\epsilon \cdot (x_t^f)^2 - x_t^f\right)\big|\mathcal{G}_{t-1} \trianglelefteq_{\eta_\epsilon} c_2 \cdot \epsilon \qquad \text{a.s. for all } \epsilon \geq 0, f \in \mathcal{F} \text{ and } t \in \mathbb{N}. \tag{17}$$

Finally, suppose that we do not start with a function $\nu$ but with a nondecreasing function $\tau : \mathbb{R}_0^+ \to \mathbb{R}_0^+$ such that $\tau(\epsilon)/\epsilon$ is nonincreasing in $\epsilon$. We can now formulate the *central condition*:

**Condition 18.** Let $\tau(\epsilon)$ be a function as above. The family (3) satisfies the $\tau$-*central condition* if there exists a constant $c > 0$ such that

$$-x_t^f \mid \mathcal{G}_{t-1} \ \trianglelefteq_{\tau(\epsilon)} \ c \cdot \epsilon \qquad \text{almost surely for all } f \in \mathcal{F}, \ \epsilon \geq 0 \text{ and } t \in \mathbb{N}. \qquad (18)$$

As promised, all four conditions are (essentially) equivalent four bounded losses, although for the proof of our main result, Theorem 2, we only needed that Bernstein with $\nu(\cdot) = (\cdot)^\kappa$ implies ESI-Bernstein. Let us briefly indicate how all the equivalences work.

1. The proof that $\nu$-Bernstein is equivalent to linearized $\nu$-Bernstein is easy, and a sketch is provided below.

2. The proof that $\nu$-ESI-Bernstein implies linearized $\nu$-Bernstein is easy: use Lemma 6 Part 1, which is just Jensen's inequality.

3. [Van Erven et al., 2015] proved that if the functions $\nu$ and $\tau$ are related by:

$$\tau(x) \cdot \nu(x) = x$$

then the $\nu$-Bernstein condition holds iff the $\tau$-central condition holds.

To complete all equivalences, it is thus sufficient to show that for any given $\tau$, $\tau$-central implies $\nu$-ESI-Bernstein, with $\nu$ related to $\tau$ as above. We clearly see that the challenge here is that superficially, $\nu$-ESI-Bernstein looks strictly stronger since the left-hand side in (17) is larger than in (18) whereas the right-hand side is the same up to a constant factor. But it turns out one can show that, with the right constant factors, the implication does hold. Our main technical lemma, Lemma 9, can be interpreted as providing just this proof, for the case that $\tau$ is a *strictly* increasing function. For then the function $\tau(\epsilon)$ has an inverse, say $\bar{\epsilon}(\eta)$, and the central condition can be restated as: there exists constant $c > 0$ such that

$$-x_t^f \mid \mathcal{G}_{t-1} \ \trianglelefteq_\eta \ c \cdot \bar{\epsilon}(\eta) \qquad \text{almost surely for all } f \in \mathcal{F}, \ \eta > 0 \text{ and } t \in \mathbb{N}, \qquad (19)$$

or equivalently,

$$\frac{1}{\eta} \ln \mathbb{E}\left[ -x_t^f \mid \mathcal{G}_{t-1} \right] \ \leq \ c \cdot \bar{\epsilon}(\eta) \qquad \text{almost surely for all } f \in \mathcal{F}, \ \eta > 0 \text{ and } t \in \mathbb{N}.$$

But this implies that the function $\bar{\epsilon}(\eta)$ is an upper bound on the normalized cumulant generating function $\epsilon(\eta)$ as defined in Section A.1. Lemma 9 then implies that

$$\left( c_1 \cdot \eta \cdot (x_t^f)^2 - x_t^f \right) \mid \mathcal{G}_{t-1} \ \trianglelefteq_\eta \ c_2 \cdot \bar{\epsilon}(\eta) \qquad \text{a.s. for all } \eta > 0, \ f \in \mathcal{F} \text{ and } t \in \mathbb{N},$$

which can be seen to be equivalent to the $\nu$-ESI condition with $\nu(y) = y/\tau(y)$, by replacing $\bar{\epsilon}(\eta)$ by $\epsilon$ and $\eta_\epsilon$ by $\tau(\epsilon)$, which can be done because $\tau$ is the inverse of $\bar{\epsilon}$.

**Proof of Equivalence $\nu$-Bernstein $\Leftrightarrow$ Linearized $\nu$-Bernstein**

*Proof.* **(sketch)** For the case that $\nu(\cdot) = (\cdot)^\kappa$ as in the original condition, the proof is straightforward and indicated above Condition 7. For the general case, consider any function $\nu$ as in the definition of the $\nu$-Bernstein condition. We will show that, for arbitrary $c_0, E > 0, F > 0$ we have:

$$c_0 F \leq \nu(E) \ \Rightarrow \ \text{for all } \epsilon > 0: \ \eta_\epsilon c_0 F - E \leq \epsilon. \qquad (20)$$

Conversely, for arbitrary $c_1, E, F > 0$ we have:

$$\text{for all } \epsilon > 0: \ \eta_\epsilon c_1 F - E \leq \epsilon \Rightarrow c_1 F \leq 2\nu(E). \qquad (21)$$

The result readily follows from these two implications by substituting, for each $t$ and $f \in \mathcal{F}$, $F = \mathbb{E}\left[ (x_t^f)^2 \big| \mathcal{G}_{t-1} \right]$, $E = \mathbb{E}\left[ x_t^f \big| \mathcal{G}_{t-1} \right]$ (and the case for $\epsilon = 0$ follows trivially by considering it separately).

To show the first implication, note that if the premise of (20) holds, then for $\epsilon \geq E$, we have $c_0 F \leq \nu(E) \leq \nu(\epsilon) = (\nu(\epsilon)/\epsilon) \cdot \epsilon = \epsilon/(\eta_\epsilon)$ and the conclusion holds. For $\epsilon < E$, we have $c_0 F \leq \nu(E) = (\nu(E)/E) \leq (\nu(\epsilon)/\epsilon)E = E/(\eta_\epsilon)$ and the conclusion holds as well.

For the second implication, first note the following equivalences:

$$\eta_\epsilon c_1 F - E \le \epsilon \Leftrightarrow \qquad c_1 F \le \frac{\epsilon + E}{\eta_\epsilon} \Leftrightarrow c_1 F \le \frac{\nu(\epsilon)}{\epsilon}(\epsilon + E) \Leftrightarrow c_1 F \le \left( \frac{\nu(\epsilon)}{\epsilon} \cdot E + \nu(\epsilon) \right)$$

Now (21) follows by noting that if its premise holds, then in particular it holds for $\epsilon = E$, and then the equivalence above gives that $c_1 F \le (\nu(E) + \nu(E))$, which was to be shown. $\qquad\square$

## C  Proof of Lemma 4

*Proof.* Since, by assumption, $u$ and $x$ have length at most 1, the hinge loss simplifies to $\ell(u) = 1 - y\langle u, x\rangle$ with gradient $\nabla \ell(u) = -yx$. This implies that

$$u^* := \arg\min_u \mathbb{E}\left[\ell(u)\right] = \frac{\mu}{\|\mu\|}, \tag{22}$$

and, for any $w$ from the unit ball,

$$
\begin{aligned}
(w - u^*)^\top \mathbb{E}\left[\nabla\ell(w)\nabla\ell(w)^\top\right](w - u^*) &= (w - u^*)^\top \mathbb{E}\left[xx^\top\right](w - u^*) \\
&\le \lambda_{\max}(w - u^*)^\top(w - u^*) \le 2\lambda_{\max}(1 - \langle w, u^*\rangle) \\
&= \frac{2\lambda_{\max}}{\|\mu\|}(w - u^*)^\top(-\mu) = \frac{2\lambda_{\max}}{\|\mu\|}(w - u^*)^\top \mathbb{E}\left[\nabla\ell(w)\right],
\end{aligned}
$$

which proves the first part of the lemma.

For the second part, we first observe that $\lambda_{\max} = 1/d$. Then, to compute $\|\mu\|$, assume without loss of generality that $\|\bar{u}\| = 1$, in which case $\bar{u} = u^*$. Now symmetry of the distribution of $x$ conditional on $\langle x, u^*\rangle$ gives

$$\mathbb{E}\left[yx \mid \langle x, u^*\rangle\right] = \text{sign}(\langle x, u^*\rangle) \mathbb{E}\left[x \mid \langle x, u^*\rangle\right] = \text{sign}(\langle x, u^*\rangle)\langle x, u^*\rangle u^* = |\langle x, u^*\rangle|u^*.$$

By rotational symmetry, we may further assume without loss of generality that $u^* = e_1$ is the first unit vector in the standard basis, and therefore

$$\|\mu\| = \|\mathbb{E}\left[|\langle x, u^*\rangle|\right]u^*\| = \mathbb{E}\left[|x_1|\right],$$

where $x_1$ is the first component of $x$. If $z = (z_1, \ldots, z_d)$ is multivariate Gaussian $\mathcal{N}(0, I)$. Then $x = z/\|z\|$ is uniformly distributed on the sphere, so

$$\mathbb{E}[|x_1|] = \mathbb{E}\left[\frac{|z_1|}{\|z\|}\right] \ge \frac{1}{4\sqrt{d}} \mathbb{P}\left(|z_1| \ge \tfrac{1}{2} \wedge \|z\| \le 2\sqrt{d}\right).$$

Since $\mathbb{P}\left(|z_1| < \tfrac{1}{2}\right) \le 0.4$ and $\mathbb{P}\left(\|z\| \ge 2\sqrt{d}\right) \le \frac{1}{4d}\mathbb{E}\left[\|z\|^2\right] = \frac{1}{4}$, we have

$$\mathbb{P}\left(|z_1| \ge \tfrac{1}{2} \wedge \|z\| \le 2\sqrt{d}\right) \ge 1 - 0.4 - \frac{1}{4} = 0.35,$$

from which the conclusion of the second part follows with $c = 8/0.35$. $\qquad\square$

## D  Continuous Models

We now consider Squint with models of predictors $\mathcal{F}$ that have uncountably many elements so that in general $\pi^{f^*} = 0$, and each $f \in \mathcal{F}$ is a function from $\mathcal{X}$ to $\mathcal{A}$, with $\ell_t^f := \ell(y_t, f(x_t))$ for some fixed loss function $\ell : \mathcal{Y} \times \mathcal{A} \to [0, 1]$. This setting includes standard parametric models in classification and regression but also countable unions thereof as well as nonparametric models. We first present an extension of Theorem 2 to this case; we then give an illustration of this result with sup-norm metric entropy numbers.

Squint can be straightforwardly applied to uncountable models, but now the weight vector $w_t$ output by Squint at time $t$ takes the form of a distribution on the set $\mathcal{F}$. For general distributions $u$ on $\mathcal{F}$, the loss that $u$ incurs at time $t$ is now defined as $\ell_t^u := \mathbb{E}_{f\sim u}[\ell_t^f]$, so that the loss of Squint at time $t$ is given by $\ell_t^{w_t}$, which generalizes the expression $\langle w_t, \ell_t\rangle$ for the countable case. The regret of Squint

relative to an arbitrary $u$ is thus given by $R_T^u = \mathbb{E}_{f \sim u} \sum_{t=1}^T (\ell_t^{w_t} - \ell_t^f)$, and the variance term in (1) generalizes to $V_T^u = \mathbb{E}_{f \sim u} \sum_{t=1}^T v_t^f$ with $v_t^f = (\ell_t^{w_t} - \ell_t^f)^2$.

For such models we will use that, as shown by Koolen [2015], Squint satisfies the following quantile or 'KL' bound:

$$R_T^u \le 2\sqrt{V_T^u K_T^u} + K_T^u \qquad (23)$$

which holds for every distribution $u$ on $\mathcal{F}$ and prior $\pi$, where now $K_T^u = O(\text{KL}(u\|\pi) + \ln\ln T)$ and $\text{KL}(u\|\pi)$ is the KL divergence between prior $\pi$ and the distribution $u$.

Note that (23) generalizes the countable bound (1), which is retrieved if $u$ is taken to be a point mass on $k$.

**Theorem 19** (Extension of Theorem 2). *In any stochastic setting satisfying the $(B, \kappa)$-Bernstein Condition 1, the guarantee (23) for Squint implies fast rates for Squint in expectation (if there is sufficient prior mass on predictors $f$ that behave similarly to $f^*$ in expectation) and with high probability (if there is sufficient prior mass on $f$ that are guaranteed to behave similarly to $f^*$ on all $x$). That is, for all $T$, for any sequence $u_1, u_2, \ldots$ of distributions on $\mathcal{F}$ and sequence of numbers $C_1, C_2, \ldots$ that satisfy*

$$\mathbb{E}\left[\sum_{t=1}^T \ell_t^{u_T}\right] \le \mathbb{E}\left[\sum_{t=1}^T \ell_t^{f^*}\right] + C_T, \qquad (24)$$

*we have*

$$\mathbb{E}[R_T^{f^*}] = O\left((K_T + C_T)^{\frac{1}{2-\kappa}} T^{\frac{1-\kappa}{2-\kappa}}\right),$$

*and if (24) holds for* every *sequence* $(x^T, y^T)$, *then we also have for any $\delta > 0$, with probability at least $1 - \delta$,*

$$R_T^{f^*} = O\left((K_T + C_T - \ln\delta)^{\frac{1}{2-\kappa}} T^{\frac{1-\kappa}{2-\kappa}}\right)$$

*where $K_T := K_T^{u_T}$ from (23).*

While this theorem does allow us to use priors $u$ with uncountable support, it is easiest to illustrate with priors with support on a discretized version (countable subset) of $\mathcal{F}$ which may assign probability 0 to $f^*$:

**Example 1.** Consider the classification setting where $\mathcal{J}$ is either finite or $\mathbb{N}$, and $\mathcal{F} = \bigcup_{j \in \mathcal{J}} \mathcal{F}_j$ is a finite or countable union of sub-models such that for $\delta > 0$, $\ddot{\mathcal{F}}_{j,\delta} \subset \mathcal{F}_j$ is a minimal $\delta$-cover of $\mathcal{F}_j$ in the $\ell_\infty$-norm (i.e. $\sup_{f \in \mathcal{F}_j} \min_{\dot{f} \in \ddot{\mathcal{F}}_{j,\delta}} \sup_{x \in \mathcal{X}, y \in \mathcal{Y}} \|\ell(y, f(x)) - \ell(y, \dot{f}(x))\| \le \delta$). Define $\Gamma := \{2^0, 2^{-1}, \ldots\}$. Assume that for all $j$, $\mathcal{N}(\mathcal{F}_j, \delta) := |\ddot{\mathcal{F}}_{j,\delta}| < \infty$ and note that $\log \mathcal{N}(\mathcal{F}_j, \delta)$ is the metric entropy of $\mathcal{F}_j$ in the sup norm at scale $\delta$. Let $\pi_{\mathcal{J}}$ be a probability mass function on $\mathcal{J}$ and let $\pi_{\mathbb{N}}$ be a probability distribution on $\mathbb{N}$ with $-\log \pi_{\mathcal{J}}(j)\pi_{\mathbb{N}}(k) = O(\log(jk))$ and let $\pi$ be the prior on $\bigcup_{j \in \mathcal{J}, \delta \in \Gamma} \ddot{F}_{j,\delta}$ with mass function $\pi$ given by, for $f \in \ddot{F}_{j,2^{-k}}$, $\pi(f) = \pi_{\mathcal{J}}(j)\pi_{\mathbb{N}}(k)/\mathcal{N}(\mathcal{F}_j, 2^{-k}))$. Then Theorem 19 gives the following bound in expectation (and mutatis mutandis in probability):

$$R_T^{f^*} = O\left(\left(\min_{j,k} T2^{-k} + \log(jk) + \log \mathcal{N}(\mathcal{F}_j, 2^{-k})\right)^{\frac{1}{2-\kappa}} T^{\frac{1-\kappa}{2-\kappa}}\right).$$

Bounds in terms of models with bounded $\ell_\infty$-entropy numbers were considered before by, e.g. Gaillard and Gerchinovitz [2015] with bounded squared error loss. We note that, if $\mathcal{F}$ has logarithmic entropy numbers (e.g. $\mathcal{F} = \mathcal{F}_1$ and $\log \mathcal{N}(\mathcal{F}_1, \epsilon) = O(-\log \epsilon)$), then, by plugging in $k = \lceil \log_2 T \rceil$, we find that this cumulative regret bound is of the form $O((\log T) \cdot T^{\frac{1-\kappa}{2-\kappa}})$, the standard rate referred to in the discussion underneath Theorem 2. In the case of larger (polynomial) entropy numbers, our bounds are presumably suboptimal compared to the bounds that can be obtained by ERM, since Squint is essentially a form of an exponentially weighted forecaster that cannot exploit the chaining technique, viz. the discussion by Gaillard and Gerchinovitz [2015], Audibert [2009] and Rakhlin and Sridharan [2014]. Nevertheless, unlike ERM, Squint is robust and will continue to achieve nontrivial regret under nonstochastic, adversarially generated data, even with polynomial entropy numbers.

In practice, one may often work with $\mathcal{F}_i$ which have small (e.g. logarithmic) entropy numbers relative to the pseudo-distance $d(f_1, f_2) = \mathbb{P}(f_1(X) \ne f_2(X))$ considered by e.g. Tsybakov [2004], Audibert [2004], which may be much smaller than the $\ell_\infty$-numbers. In such cases, Theorem 19 can still be used to give good bounds in expectation.

**Proof of Theorem 19** Consider a (for now) arbitrary sequence $u_1, u_2, \ldots$, define $K_T := K_T^{u_T}$ and $K_T' = K_T/4$, and

$$C_T' = -(R_T^{u_T} - R_T^{f^*}) \text{ or equivalently } \sum_{t=1}^T \ell_t^{u_T} = \sum_{t=1}^T \ell_t^{f^*} + C_T'.$$

One easily shows that for general $a, b, c \in \mathbb{R}$, one has $(a-b)^2/2 \le (a-c)^2 + (b-c)^2$. Applying the statement with $a = \ell_t^{w_t}$, $b = \ell_t^f$ and $c = \ell^{f_t^*}$ gives $v_t^f \le 2(v_t^{f^*} + (x_t^f)^2)$. Summing over $t = 1..T$ and taking expectation over $f \sim u_T$ now gives $V_T^{u_T} \le 2V_T^{f^*} + 2E_T$ where $E_T = \mathbb{E}_{f \sim u_T}\left[\sum_{t=1}^T (x_t^f)^2\right]$.

Applying this to the bound (23) above at $u_T$, we get

$$R_T^{f^*} \le C_T' + 2\sqrt{(V_T^{f^*} + E_T)2K_T'} + K_T' = \inf_\eta\left\{C_T' + \eta(V_T^{f^*} + E_T) + \frac{2K_T'}{\eta} + K_T'\right\}. \tag{25}$$

We first prove an analogue to Theorem 11 for the uncountable setting, based on (25). As in that theorem, let, for given $\mathcal{F}$, $\left\{x_t^f \big| f \in \mathcal{F}\right\}$ be the associated the excess loss family from (3), and let $\epsilon(\eta)$ be, as in (9), the corresponding maximal normalized cumulant generating function. Let $K_T'$ be as above. Fix $\gamma \ge 0$ and let $c$ be as in Lemma 9. Now as in the proof of Theorem 11 we have for all $f \in \mathcal{F}$, $x_t^f \in [-1, 1]$ and $-x_t^f \unlhd_\eta \epsilon(\eta)$ by construction of $\epsilon(\eta)$. Hence $-x_t^f \unlhd_\gamma \epsilon(\gamma)$ for all $f \in \mathcal{F}$, which implies $-\mathbb{E}_{f \sim w_t} x_t^f \unlhd_\gamma \epsilon(\gamma)$ and also $-\mathbb{E}_{f \sim u_T} x_t^f \unlhd_\gamma \epsilon(\gamma)$, and hence by Lemma 9 (see remark below the lemma),

$$c\gamma \underset{f \sim w_t}{\mathbb{E}}[x_t^f]^2 - \underset{f \sim w_t}{\mathbb{E}}[x_t^f] \unlhd_\gamma \epsilon(2\gamma)(1 + c\gamma^2) \text{ and} \tag{26}$$

$$c\gamma \underset{f \sim u_T}{\mathbb{E}}[x_t^f]^2 - \underset{f \sim u_T}{\mathbb{E}}[x_t^f] \unlhd_\gamma \epsilon(2\gamma)(1 + c\gamma^2). \tag{27}$$

Using $r_t^{f^*} = \mathbb{E}_{f \sim w_t}\left[x_t^f\right]$, again analogously to the proof of Theorem 11, we may telescope (26) over rounds to

$$c\gamma V_T^{f^*} - R_T^{f^*} \unlhd_\gamma T\epsilon(2\gamma)(1 + c\gamma^2) \tag{28}$$

Now we use (25) with $\eta = \frac{c\gamma}{2}$, which implies $2R_T^{f^*} \le 2C_T' + c\gamma(V_T^{f^*} + E_T) + 4K_T'/(c\gamma) + 2K_T'$. Combining this with (28), we find:

$$U \unlhd 0 \text{ with } U = \gamma R_T^{f^*} - \left(2\gamma C_T' + c\gamma^2 E_T + \frac{4K_T'}{c} + \gamma 2K_T' + \gamma T\epsilon(2\gamma)(1 + c\gamma^2)\right). \tag{29}$$

Similarly to deriving (28), using the definition of $E_T$, we may telescope (27) over rounds to get

$$U' \unlhd 0 \text{ with } U' = c\gamma^2 E_T - \gamma C_T' - \gamma T\epsilon(2\gamma)(1 + c\gamma^2). \tag{30}$$

We may now combine (29) and (30) using Lemma 6 with $w$ a distribution that puts mass $1/2$ on random variable $U$ and $1/2$ on $U'$, to get $(U + U')/2 \unlhd 0$, which can be rewritten to:

$$\frac{\gamma}{2}\left(R_T^{f^*} - (2C_T' + c\gamma \frac{E}{T} + \frac{4K_T'}{c\gamma} + 2K_T' + T\epsilon(2\gamma)(1 + c\gamma^2)) + c\gamma E_T - \gamma C_T' - \gamma\epsilon(2\gamma)(1 + c\gamma^2)\right)$$
$$\unlhd 0,$$

and further to

$$\frac{1}{2}R_T^{f^*} \unlhd_\gamma \frac{3}{2}C_T' + \frac{K_T}{c\gamma} + T\epsilon(2\gamma)(1 + c\gamma^2) + 2K_T', \tag{31}$$

which is the required analogue of the statement of Theorem 11. Note that this statement holds for *every* sequence $u_1, \ldots, u_T$, and $C_T'$ is a random variable that depends on data $(x^T, y^T)$.

The remainder of the proof of Theorem 19 now follows in a fashion entirely analogous to the proof of Theorem 2, as in Appendix A.3, where we use that we can bound $C_T'$ by $C_T$, either in expectation or on all sequences; we omit further details.