[Reviews · NeurIPS 2016]

Reviewer 1

Summary

The paper continues a line of research getting more and more popular in online learning (and learning theory in general): how to design algorithms that perform well in the adversarial setting yet are able to utilize additional regularities in the data that make the task at hand easy. It is shown that some existing algorithms in online learning (Squint and MetaGrad) are able to adapt to easy stochastic environments satisfying the Bernstein condition (used in proving fast rates in stochastic loss minimization), while keeping the usual adversarial guarantees.

Qualitative Assessment

ADDED AFTER THE RESPONSE: I am happy with the response and support the acceptance of the paper. Please include the discussions from the rebuttal to the paper. Two minor things: - I still think that unifying the notation is possible (in the expert case you have to restrict U to the corners), but I might be missing something here. - Bernstein to Central: Application of the newly cited lemma is ok, but mention the required boundedness conditions (|2\eta x^f| \le 1.79... which requires D,G-dependent assumptions on eta for the OCO case. ============================================== In general I find the problem important and interesting and the treatment elegant. The derivation of the results do not seem to require new techniques, although certainly there are some interesting technicalities in the proofs. While the presentation is good, it could be improved: I think that related work should be discussed a bit differently, and the organization of the paper can be improved to put the results more in context and emphasize on the more relevant parts (these can be done easily, suggestions are given below). First of all, a slightly weakened version of the intended adaptive bounds can be achieved trivially by applying either the AB-prod algorithm of Sani et al (2014) or the algorithm of the much earlier paper of Even-Dar et al (Machine Learning 2008, "Regret to the best vs. regret to the average, this should also be cited) for an adversarial learning algorithm and the an algorithm that can adapt to the Bernstein condition in the stochastic case (e.g., Bartlet and Mendelson, 2006). You should argue how and why your results are better than this (which is not hard). (This argument also applies as a solution to the "linear worst case regret" in l. 169.) The results essentially apply to Squint and MetaGrad, but these algorithms are not provided. While Squint is at least published, MetaGrad is only available as a preprint (and by no means is so widely known that it should not be included). The paper continuously derives results for Squint and MetaGrad separately, from time to time introducing some unifying notation. It seems to me that treating the expert case (Hedge setting) as a problem of linear prediction over the simplex would automatically unify the two problems (the original losses will become gradients). In the statements of the main results (Theorem, 3 Theorem 9), why do you recall MetaGrad and Squint? Wouldn't the main statement be that given data dependent regret bounds fast rates hold for the Bernstein case? By the way, wouldn't your results apply for Adapt-ML-Prod (maybe I am missing something here)? * Organization: I think that Lemma 7 (and the related notation) adds very little to understanding the proof of the theorem. Especially, the trivial proofs should be relegated to the appendix. This could free some space to provide more important materials (e.g., more discussion, high level proof steps of Theorem 9, or the proof of Theorem 3). * Technical questions: 1. In the problem definition, make it clear if the environment is oblivious or not. While in general your results apply to non-oblivious environments, conditional expectations (e.g., in Sec 2.3) are not taken according to that. 2. Sec 2.3: Wouldn't it be enough to require that k^* minimizes the expected cumulative loss? 3. Since the Hedge setting is only defined for countably infinite expert sets, it is somewhat bothering to extend it to the undefined uncountable case. Maybe define the problem generally, then restrict it to the countable case when necessary? 4. Appendix B: It would be better to repeat the Bernstein Sandwich lemma (it is very hard to resolve the notation in the reference). Also, I am not sure that the cited lemma is correct: in the last step of its proof, both sides are multiplied by x^2, but why can you take its minimum in the upper bound? (There are too many parentheses in line 424.) 5. Theorem 13: I find the statement of the theorem a bit confusing. In particular, the O(K_T+C_T) notation is a bit fuzzy when nothing is assumed about these sequences. Why not make an assumption that K_T/T and C_T/T go to zero, in which case the bound would have the nice form of T O(((K_T+C_T)/T)^1/(2-kappa)). Also consider removing the intuitive explanations from mathematical statements (maybe this is just my taste). It is very hard to associate any meaning to this part of the theorem. * Minor comments: - Introduction, par 2: There are several other types of results (combinatorial problems, context tree prediction, tracking, curvature of the loss function, etc) which adapt to some sort of problem complexity. Consider mentioning what sort of complexity you are after. - l. 66: The definition of \pi might not be entirely clear, in particular that the upper index is not power and that the distribution is over the natural numbers (it will be automatically if you define the algorithm). - l. 100: What do you mean by "the Bays act is in the model?" - l. 116: "these stochastic conditions" -> the following stochastic conditions. - l. 122: only equivalent with appropriate choices of parameters. - l. 130: "Second order methods" seems to be a bit too general here. - l. 175-180 is not entirely clear (without reading the relevant part in the Appendix). Why is this discussion restricted to Squint? - l. 191-193: In fact you loose a log log T factor by relaxing the assumptions - l. 198: consider->considered - l. 218: "we model" should rather be "Assume z_1,...,z_T is a Markov chain" - Section 3.4: I find it a bit overstatement to call the considered linear loss Hinge loss (since the "hinge part" is excluded). In what sense is the maximum of the Hinge loss 1 (consider a large negative < x,u >)? - Lemma 7: Your statement in 1 is really not that X is negative with high probability, only that it is not too large. Also, the notation about the expectations is very confusing here (e.g., E_{f\sim w} only takes expectation over f (with all other randomness in X^f fixed, wile E does not affect f in E X^f. Please make these more precise/clear (define them; applies to other occurrences, e.g., the proof of Theorem 9). - l. 284: give a reference for the dual representation. - Theorem 9: remove "the" before "excess". gamma is not defined to be 2 eta. This is also needed in the proof (l. 434). - l. 312: 2 epsilon should be 2 gamma - The references are really hard to parse in the current format. - l. 435: You have to take the proper conditional expectations to be able to invoke the chain rule. - l. 455: Mention that the omitted kappa-dependent constant is at most 1. - Example 1: What is \mathcal{J}? Please make the notation consistent throughout. (correct "m utandis") - l. 459: Did you use an assumption that \gamma^2 \le 1 (to deal with (the 1+gamma^2) terms)?

Confidence in this Review

3-Expert (read the paper in detail, know the area, quite certain of my opinion)


Reviewer 2

Summary

The authors show that algorithms whose regret is upper bounded by a second order variance based bounds, achieve very fast convergence rates in stochastic environments that are "favorable" in the sense that they satisfy the previously introduced Bernstein condition for some kappa. For kappa=0 this coincides with the fully adversarial setting, while for kappa=1, it coincides with a generalization of the stochastic i.i.d. setting with a bounded gap between the best and the second best expert. They show that the second order bound implies directly a bound which becomes polylogaritmic in T for kappa=1, while it goes smoothly to \sqrt{T} as kappa goes to 0. Thereby giving a nice interpolation of the regret rate of the algorithm from nice to worst case environments. Such variance based bounds are achieved by existing algorithms such as Squint and MetaGrad. Their analysis applies to the expert setting as well as to the more general online convex optimization setting. The authors complement their analysis with several nice examples of how their approach yields fast convergence rates in specific settings.

Qualitative Assessment

I found the paper very interesting. I think it very nicely combines the machinery of the Bernstein condition and the Central condition that was used in previous papers to show faster convergence in statistical learning settings, together with existing regret bounds of online learning algorithms with respect to the second order terms, to give the best of both worlds. For that reason I recommend acceptance.

Confidence in this Review

1-Less confident (might not have understood significant parts)


Reviewer 3

Summary

This paper shows that a particular form of second order regret bound for both the Hedge setting and the more general OCO setting can automatically adapt to the "friendliness" of stochastic environments characterized by the Bernstein condition and lead to fast rates both in expectation and with high probability. Several examples are discussed. The fast rates for some of these examples appear to be unknown previously.

Qualitative Assessment

Overall the results in this paper are interesting, especially for the fast rates in some examples such as OCO with absolute loss. However, the paper is not very well written in my opinion, especially for the later parts. Some questions and suggestions are listed below: 1. It seems to me that the fast rate (at least for the expectation bound) can be derived directly by applying Condition 1 to the second order bound using Jensen’s inequality, but the paper provides a more complicated proof through Condition 2. Is it all because that’s necessary for getting the high probability bound? 2. I am not sure I understand Section 4.2 and 4.3 and their connections to the main proof. I got lost when reading Line 282-285 and 287-290. What is the point of these discussions? And how does that connect to Appendix B? I thought Appendix B (which was already proven by Van Erven el al. 2015?) was enough for the whole arguments. Same issue for Section 4.3, what does “x is special to degree \epsilon(\eta)” mean? 3. In general I wound suggest discussing those interesting examples in more details, such as moving some of the contents in Appendix F to the main text and shorten Sec 4. As it’s written, I can’t tell which part of the analysis is novel and important. 4. Some detailed comments: 1) Line 37, “a variety of”. I thought the paper only discussed one specific second order bound? 2) The term K_T in the regret causes some confusion since the number of experts is denoted by K also (Line 69). 3) Line 94: “loss a fixed one-dimensional convex loss function”, but it has two arguments? 4) Line 101: best -> the best 5) Line 122: “Bernstein condition is equivalent to…”, shouldn’t that depend on what \epsilon is? 6) I don’t understand the argument in Line 211-214. Maybe make it more formal and clear? 7) Sec 3.4, X is not defined. 8) Line 312: \epsilon(2\epsilon) -> \epsilon(2\gamma) ?

Confidence in this Review

3-Expert (read the paper in detail, know the area, quite certain of my opinion)


Reviewer 4

Summary

The paper provides online learning algorithms with worst case regret guarantee in the adversarial setting but tighter guarantees in favorable stochastic setting, for well known experts and online convex optimization problems. The "favorability" of the stochastic setting is defined through a Bernstein condition. The paper describes the regret guarantees of Squint (for experts) and Metagrad (for OCO) in the adversarial setting and then shows how, for various examples, the Bernstein condition is met, along with faster regret rates.

Qualitative Assessment

I found the paper (mostly) clearly written and felt that it has strong technical content. I think I mostly understood what the authors were getting at, but I would like the authors to address couple of issues which I found confusing. 1. Since I am not familiar with Squint and Metagrad algorithms, what I could not figure out from Eq. (1) and (2) is: are the algorithms some form of meta algorithms? That is, since the quantity V_k in both (1) and (2) simply contains the predictions (w_t s), is it that Squint (and Metagrad) uses some type of underlying sub-routine algorithm, which would give a bound on V_k and hence we would get the overall regret? The reason I am asking this is that in Theorem 3, the regret guarantee is give in term of time horizon T, as long as the stochastic setting meets Bernstein condition. But the Squint and Metagrad algorithm does not have T explicitly in the regret bound. 2. In the Hedge setting with gap \alpha, why doesn't the regret bound have the gap \alpha and number of experts K (assuming that there are finite number of experts K). For eg., to the best of my knowledge, the hedge algorithm for K experts problem gives a regret guarantee of \sqrt{T \log(K)}, without making any assumption on the generation process of the experts' losses. Now, in the same setting (K experts), if we assume that the losses are i.i.d distributed with gap \alpha, should we not have a regret bound which has the terms \alpha and K? Following Theorem 3, the regret rate is just posed as O( \log \log T), without \alpha and K.

Confidence in this Review

1-Less confident (might not have understood significant parts)


Reviewer 5

Summary

The paper investigates conditions in online learning under which some state-of-the-art algorithms exhibit not only robust and safe performance in worst case (i.e. square-root regret), but also small (logarithmic) regret in "friendly" distributions of data. In particular the paper focuses on MetaGrad [Koolen and van Erven, 2016] and Squint [Koolen and van Erven, 2015] in Online Convex Optimization (OCO) setting and Hedge setting, respectively. The "friendliness" of distribution of data is quantified by Bernstein condition which controls the variance of excess loss of the predictions compared to the best prediction/expert. Furthermore, the paper also demonstrates some examples (including well-known ones) satisfying Bernstein conditions, and consequently, enjoying both adversarial guarantees and stochastic fast rates.

Qualitative Assessment

The paper is farily well-presented. In addition, the analysis provided by the paper seems sound. The paper provides the regret bound results not only in-expectation (similar to [Koolen and van Erven, 2016]) but also in-probability.

Confidence in this Review

1-Less confident (might not have understood significant parts)


Reviewer 6

Summary

The authors study achievable regret bounds in online convex optimization and in the hedge setting. In this line of research, it is well known that, when the loss functions are adversarially chosen, rates faster than O(\sqrt(T)) are in general not possible without making more assumptions on the loss functions or on the decision sets. The authors establish that two algorithms developed in the literature, namely MetaGrad and Squint, achieve faster rates when some precise stochastic assumptions on the loss functions characterizing the difficulty of the underlying learning problem are satisfied while still guaranteeing O(\sqrt(T)) regret bounds if the loss functions happen to be adversarially chosen. The authors give a number of practical examples where these assumptions can be shown to hold.

Qualitative Assessment

All results are properly substantiated with proof, the manuscript is well written, the results are well presented and illustrated with many examples, and the proofs are easy to follow. The topic has attracted a lot of attention in recent years in the online learning community (e.g. "Optimization, learning, and games with predictable sequences" by S. Rakhlin and K. Sridharan) and in general it is hard to find algorithms that guarantee both asymptotically-optimal worst-case regret bounds and faster rates when the loss functions are stochastic so I really enjoyed reading the paper. My main concerns are: (i) this could be considered only a slight incremental progress on the topic over the work of Gaillard et al. [2014], (ii) the authors do not compare their results to the ones obtained in a recently published paper "Fast Rates in Statistical and Online Learning" by T. Ervan et al., and (iii) the example given Section 3.2 is quite artificial.

Confidence in this Review

2-Confident (read it all; understood it all reasonably well)


Reviewer 7

Summary

This paper considers online learning algorithms that guarantee worst case regret rates in adversarial environments and adopt optimally to favorable stochastic environments. The friendliness of the stochastic setting is quantified with the Bernstein condition. Two recent algorithms are shown to adapt automatically to the Bernstein parameters of the stochastic case.

Qualitative Assessment

The problem in consideration is interesting. The results also seem to be solid. The authors could provide more explanations for the technical results. 1. It would be nice to have some explanations about Conditions 1 and 2. In particular, how general are the two conditions? 2. It seems that the results in the paper are derived for algorithms that can guarantee (1) and (2). It would be nice if the authors could discuss more about how the results can also facilitate algorithm design for problems of this type.

Confidence in this Review

1-Less confident (might not have understood significant parts)